# META-Q-LEARNING

**Rasool Fakoor[1], Pratik Chaudhari[2][*] Stefano Soatto[1], Alexander Smola[1]**
[1] Amazon Web Services
[2] University of Pennsylvania
Email: {fakoor, soattos, smola}@amazon.com, pratikac@seas.upenn.edu

## ABSTRACT

This paper introduces Meta-Q-Learning (MQL), a new off-policy algorithm for meta-Reinforcement Learning (meta-RL). MQL builds upon three simple ideas. First, we show that Q-learning is competitive with state-of-the-art meta-RL algorithms if given access to a context variable that is a representation of the past trajectory. Second, a multi-task objective to maximize the average reward across the training tasks is an effective method to meta-train RL policies. Third, past data from the meta-training replay buffer can be recycled to adapt the policy on a new task using off-policy updates. MQL draws upon ideas in propensity estimation to do so and thereby amplifies the amount of available data for adaptation. Experiments on standard continuous-control benchmarks suggest that MQL compares favorably with the state of the art in meta-RL.

## 1 INTRODUCTION

Reinforcement Learning (RL) algorithms have demonstrated good performance on simulated data. There are however two main challenges in translating this performance to real robots: (i) robots are complex and fragile which precludes extensive data collection, and (ii) a real robot may face an environment that is different than the simulated environment it was trained in. This has fueled research into Meta-Reinforcement Learning (meta-RL) which develops algorithms that "meta-train" on a large number of different environments, e.g., simulated ones, and aim to adapt to a new environment with few data.

How well does meta-RL work today? Fig. 1 shows the performance of two prototypical meta-RL algorithms on four standard continuous-control benchmarks.[1] We compared them to the following simple baseline: an off-policy RL algorithm (TD3 by Fujimoto et al. (2018b)) and which was trained to maximize the average reward over all training tasks and modified to use a "context variable" that represents the trajectory. All algorithms in this figure use the same evaluation protocol. It is surprising that this

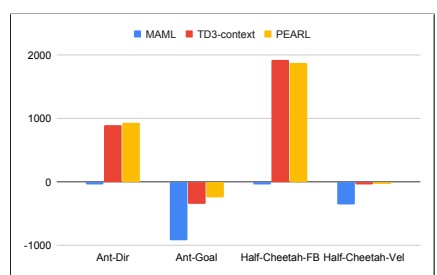

Figure 1: **How well does meta-RL work?** Average returns on validation tasks compared for two prototypical meta-RL algorithms, MAML (Finn et al., 2017) and PEARL (Rakelly et al., 2019), with those of a vanilla Q-learning algorithm named TD3 (Fujimoto et al., 2018b) that was modified to incorporate a context variable that is a representation of the trajectory from a task (TD3-context). Even without any meta-training and adaptation on a new task, TD3-context is competitive with these sophisticated algorithms.

simple non-meta-learning-based method is competitive with state-of-the-art meta-RL algorithms. This is the first contribution of our paper: we demonstrate that it is not necessary to meta-train policies to do well on existing benchmarks.

Our second contribution is an off-policy meta-RL algorithm named Meta-Q-Learning (MQL) that builds upon the above result. MQL uses a simple meta-training procedure: it maximizes the average

---

[*]Work done while at Amazon Web Services
[1]We obtained the numbers for MAML and PEARL from training logs published by Rakelly et al. (2019).

rewards across all meta-training tasks using off-policy updates to obtain

$$\widehat{\theta}_{\text{meta}} = \arg\max_\theta \frac{1}{n} \sum_{k=1}^n \mathop{\mathbb{E}}_{\tau \sim D^k} \left[ \ell^k(\theta) \right] \tag{1}$$

where $\ell^k(\theta)$ is the objective evaluated on the transition $\tau$ obtained from the task $D^k(\theta)$, e.g., 1-step temporal-difference (TD) error would set $\ell^k(\theta) = \text{TD}^2(\theta; \tau)$. This objective, which we call the multi-task objective, is the simplest form of meta-training.

For adapting the policy to a new task, MQL samples transitions from the meta-training replay buffer that are similar to those from the new task. This amplifies the amount of data available for adaptation but it is difficult to do because of the large potential bias. We use techniques from the propensity estimation literature for performing this adaptation and the off-policy updates of MQL are crucial to doing so. The adaptation phase of MQL solves

$$\arg\max_\theta \left\{ \mathop{\mathbb{E}}_{\tau \sim D^{\text{new}}} \left[ \ell^{\text{new}}(\theta) \right] + \mathop{\mathbb{E}}_{\tau \sim \mathcal{D}_{\text{meta}}} \left[ \beta(\tau; D^{\text{new}}, \mathcal{D}_{\text{meta}}) \, \ell^{\text{new}}(\theta) \right] - \left( 1 - \widehat{\text{ESS}} \right) \| \theta - \widehat{\theta}_{\text{meta}} \|_2^2 \right\} \tag{2}$$

where $\mathcal{D}_{\text{meta}}$ is the meta-training replay buffer, the propensity score $\beta(\tau; D^{\text{new}}, \mathcal{D}_{\text{meta}})$ is the odds of a transition $\tau$ belonging to $D^{\text{new}}$ versus $\mathcal{D}_{\text{meta}}$, and $\widehat{\text{ESS}}$ is the Effective Sample Size between $D^{\text{new}}$ and $\mathcal{D}_{\text{meta}}$ that is a measure of the similarly of the new task with the meta-training tasks. The first term computes off-policy updates on the new task, the second term performs $\beta(\cdot)$-weighted off-policy updates on old data, while the third term is an automatically adapting proximal term that prevents degradation of the policy during adaptation.

We perform extensive experiments in Sec. 4.2 including ablation studies using standard meta-RL benchmarks that demonstrate that MQL policies obtain higher average returns on new tasks even if they are meta-trained for fewer time-steps than state-of-the-art algorithms.

## 2 BACKGROUND

This section introduces notation and formalizes the meta-RL problem. We discuss techniques for estimating the importance ratio between two probability distributions in Sec. 2.2.

Consider a Markov Decision Processes (MDP) denoted by

$$x_{t+1} = f^k(x_t, u_t, \xi_t) \quad x_0 \sim p_0^k, \tag{3}$$

where $x_t \in X \subset \mathbb{R}^d$ are the states and $u_t \in U \subset \mathbb{R}^p$ are the actions. The dynamics $f^k$ is parameterized by $k \in \{1, \ldots, n\}$ where each $k$ corresponds to a different task. The domain of all these tasks, $X$ for the states and $U$ for the actions, is the same. The distribution $p_0^k$ denotes the initial state distribution and $\xi_t$ is the noise in the dynamics. Given a deterministic policy $u_\theta(x_t)$, the action-value function for $\gamma$-discounted future rewards $r_t^k := r^k(x_t, u_\theta(x_t))$ over an infinite time-horizon is

$$q^k(x, u) = \mathop{\mathbb{E}}_{\xi_{(\cdot)}} \left[ \sum_{t=0}^\infty \gamma^t \, r_t^k \mid x_0 = x, u_0 = u, u_t = u_\theta(x_t) \right]. \tag{4}$$

Note that we have assumed that different tasks have the same state and action space and may only differ in their dynamics $f^k$ and reward function $r^k$. Given one task $k \in \{1, \ldots, n\}$, the standard Reinforcement Learning (RL) formalism solves for

$$\widehat{\theta^k} = \arg\max_\theta \ell^k(\theta) \quad \text{where } \ell^k(\theta) = \mathop{\mathbb{E}}_{x \sim p_0} \left[ q^k(x, u_\theta(x)) \right]. \tag{5}$$

Let us denote the dataset of all states, actions and rewards pertaining to a task $k$ and policy $u_\theta(x)$ by

$$D^k(\theta) = \left\{ x_t, u_\theta(x_t), r^k, x_{t+1} = f^k(x_t, u_\theta(x_t), \xi_t) \right\}_{t \geq 0, \, x(0) \sim p_0^k, \, \xi_{(\cdot)}};$$

we will often refer to $D^k$ as the "task" itself. The Deterministic Policy Gradient (DPG) algorithm (Silver et al., 2014) for solving (5) learns a $\varphi$-parameterized approximation $q_\varphi$ to the optimal value func-

tion $q^k$ by minimizing the Bellman error and the optimal policy $u_\theta$ that maximizes this approximation by solving the coupled optimization problem

$$
\begin{aligned}
\widehat{\varphi^k} &= \arg\min_\varphi \mathbb{E}_{\tau \sim D^k} \left[ \left( q_\varphi(x, u) - r^k - \gamma\, q_\varphi(x', u_{\widehat{\theta^k}}(x')) \right)^2 \right], \\
\widehat{\theta^k} &= \arg\max_\theta \mathbb{E}_{\tau \sim D^k} \left[ q_{\widehat{\varphi^k}}(x, u_\theta(x)) \right].
\end{aligned}
\tag{6}
$$

The 1-step temporal difference error (TD error) is defined as

$$
\text{TD}^2(\theta) = \left( q_\varphi(x, u) - r^k - \gamma\, q_\varphi(x', u_\theta(x')) \right)^2
\tag{7}
$$

where we keep the dependence of $\text{TD}(\cdot)$ on $\varphi$ implicit. DPG, or its deep network-based variant DDPG (Lillicrap et al., 2015), is an off-policy algorithm. This means that the expectations in (6) are computed using data that need not be generated by the policy being optimized ($u_\theta$), this data can come from some other policy.

In the sequel, we will focus on the parameters $\theta$ parameterizing the policy. The parameters $\varphi$ of the value function are always updated to minimize the TD-error and are omitted for clarity.

## 2.1 META-REINFORCEMENT LEARNING (META-RL)

Meta-RL is a technique to learn an inductive bias that accelerates the learning of a new task by training on a large of number of training tasks. Formally, meta-training on tasks from the meta-training set $\mathcal{D}_{\text{meta}} = \left\{ D^k \right\}_{k=1,\dots,n}$ involves learning a policy

$$
\widehat{\theta}_{\text{meta}} = \arg\max_\theta \frac{1}{n} \sum_{k=1}^n \ell_{\text{meta}}^k(\theta)
\tag{8}
$$

where $\ell_{\text{meta}}^k(\theta)$ is a meta-training loss that depends on the particular method. Gradient-based meta-RL, let us take MAML by Finn et al. (2017) as a concrete example, sets

$$
\ell_{\text{meta}}^k(\theta) = \ell^k(\theta + \alpha \nabla_\theta \ell^k(\theta))
\tag{9}
$$

for a step-size $\alpha > 0$; $\ell^k(\theta)$ is the objective of non-meta-RL (5). In this case $\ell_{\text{meta}}^k$ is the objective obtained on the task $D^k$ after one (or in general, more) updates of the policy on the task. The idea behind this is that even if the policy $\widehat{\theta}_{\text{meta}}$ does not perform well on all tasks in $\mathcal{D}_{\text{meta}}$ it may be updated quickly on a new task $D^{\text{new}}$ to obtain a well-performing policy. This can either be done using the same procedure as that of meta-training time, i.e., by maximizing $\ell_{\text{meta}}^{\text{new}}(\theta)$ with the policy $\widehat{\theta}_{\text{meta}}$ as the initialization, or by some other adaptation procedure. The meta-training method and the adaptation method in meta-RL, and meta-learning in general, can be different from each other.

## 2.2 LOGISTIC REGRESSION FOR ESTIMATING THE PROPENSITY SCORE

Consider standard supervised learning: given two distributions $q(x)$ (say, train) and $p(x)$ (say, test), we would like to estimate how a model's predictions $\hat{y}(x)$ change across them. This is formally done using importance sampling:

$$
\mathbb{E}_{x \sim p(x)} \mathbb{E}_{y|x} \left[ \ell(y, \hat{y}(x)) \right] = \mathbb{E}_{x \sim q(x)} \mathbb{E}_{y|x} \left[ \beta(x)\, \ell(y, \hat{y}(x)) \right];
\tag{10}
$$

where $y|x$ are the true labels of data, the predictions of the model are $\hat{y}(x)$ and $\ell(y, \hat{y}(x))$ is the loss for each datum $(x, y)$. The importance ratio $\beta(x) = \frac{dp}{dq}(x)$, also known as the propensity score, is the Radon-Nikodym derivative (Resnick, 2013) of the two data densities and measures the odds of a sample $x$ coming from the distribution $p$ versus the distribution $q$. In practice, we do not know the densities $q(x)$ and $p(x)$ and therefore need to estimate $\beta(x)$ using some finite data $X_q = \{x_1, \dots, x_m\}$ drawn from $q$ and $X_p = \{x'_1, \dots, x'_m\}$ drawn from $p$. As Agarwal et al. (2011) show, this is easy to do using logistic regression. Set $z_k = 1$ to be the labels for the data in $X_q$ and $z_k = -1$ to be the labels of the data in $X_p$ for $k \leq m$ and fit a logistic classifier on the combined $2m$

samples by solving

$$w^* = \min_w \ \frac{1}{2m} \sum_{(x,z)} \log\left(1 + e^{-zw^\top x}\right) + c\,\|w\|^2. \tag{11}$$

This gives

$$\beta(x) = \frac{\mathbb{P}(z=-1|x)}{\mathbb{P}(z=1|x)} = e^{-w^{*\top}x}. \tag{12}$$

**Normalized Effective Sample Size ($\widehat{\text{ESS}}$):** A related quantity to $\beta(x)$ is the normalized Effective Sample Size ($\widehat{\text{ESS}}$) which we define as the relative number of samples from the target distribution $p(x)$ required to obtain an estimator with performance (say, variance) equal to that of the importance sampling estimator (10). It is not possible to compute the $\widehat{\text{ESS}}$ without knowing both densities $q(x)$ and $p(x)$ but there are many heuristics for estimating it. A popular one in the Monte Carlo literature (Kong, 1992; Smith, 2013; Elvira et al., 2018) is

$$\widehat{\text{ESS}} = \frac{1}{m}\,\frac{\left(\sum_{k=1}^m \beta(x_k)\right)^2}{\sum_{k=1}^m \beta(x_k)^2} \in [0,1] \tag{13}$$

where $X = \{x_1, \ldots, x_m\}$ is some finite batch of data. Observe that if two distributions $q$ and $p$ are close then the $\widehat{\text{ESS}}$ is close to one; if they are far apart the $\widehat{\text{ESS}}$ is close to zero.

## 3  MQL

This section describes the MQL algorithm. We begin by describing the meta-training procedure of MQL including a discussion of multi-task training in Sec. 3.1. The adaptation procedure is described in Sec. 3.2.

### 3.1  META-TRAINING

MQL performs meta-training using the multi-task objective. Note that if one sets

$$\ell_{\text{meta}}^k(\theta) \triangleq \ell^k(\theta) = \mathop{\mathbb{E}}_{x \sim p_0^k}\left[q^k(x, u_\theta(x))\right] \tag{14}$$

in (8) then the parameters $\widehat{\theta}_{\text{meta}}$ are such that they maximize the average returns over all tasks from the meta-training set. We use an off-policy algorithm named TD3 (Fujimoto et al., 2018b) as the building block and solve for

$$\widehat{\theta}_{\text{meta}} = \arg\min_\theta \frac{1}{n}\sum_{k=1}^n \mathop{\mathbb{E}}_{\tau \sim D^k}\left[\text{TD}^2(\theta)\right]; \tag{15}$$

where $\text{TD}(\cdot)$ is defined in (7). As is standard in TD3, we use two action-value functions parameterized by $\varphi_1$ and $\varphi_2$ and take their minimum to compute the target in (7). This trick known as "double-Q-learning" reduces the over-estimation bias. Let us emphasize that (14) is a special case of the procedure outlined in (8). The following remark explains why MQL uses the multi-task objective as opposed to the meta-training objective used, for instance, in existing gradient-based meta-RL algorithms.

**Remark 1.** Let us compare the critical points of the $m$-step MAML objective (9) to those of the multi-task objective which uses (14). As is done by the authors in Nichol et al. (2018), we can perform a Taylor series expansion around the parameters $\theta$ to obtain

$$\nabla \ell_{\text{meta}}^k(\theta) = \nabla \ell^k(\theta) + 2\alpha(m-1)\left(\nabla^2 \ell^k(\theta)\right)\nabla \ell^k(\theta) + \mathcal{O}(\alpha^2). \tag{16}$$

Further, note that $\nabla \ell_{\text{meta}}^k$ in (16) is also the gradient of the loss

$$\ell^k(\theta) + \alpha(m-1)\|\nabla \ell^k(\theta)\|_2^2 \tag{17}$$

up to first order. This lends a new interpretation that MAML is attracted towards regions in the loss landscape that under-fit on individual tasks: parameters with large $\|\nabla \ell^k\|_2$ will be far from the local maxima of $\ell^k(\theta)$. The parameters $\alpha$ and $m$ control this under-fitting. Larger the number of gradient steps, larger the under-fitting effect. This remark suggests that the adaptation speed of gradient-based meta-learning comes at the cost of under-fitting on the tasks.

### 3.1.1 DESIGNING CONTEXT

As discussed in Sec. 1 and 4.4, the identity of the task in meta-RL can be thought of as the hidden variable of an underlying partially-observable MDP. The optimal policy on the entire trajectory of the states, actions and the rewards. We therefore design a recurrent context variable $z_t$ that depends on $\{(x_i, u_i, r_i)\}_{i \leq t}$. We set $z_t$ to the hidden state at time $t$ of a Gated Recurrent Unit (GRU by Cho et al. (2014)) model. All the policies $u_\theta(x)$ and value functions $q_\varphi(x, u)$ in MQL are conditioned on the context and implemented as $u_\theta(x, z)$ and $q_\varphi(x, u, z)$. Any other recurrent model can be used to design the context; we used a GRU because it offers a good trade-off between a rich representation and computational complexity.

**Remark 2 (MQL uses a deterministic context that is not permutation invariant).** We have aimed for simplicity while designing the context. The context in MQL is built using an off-the-shelf model like GRU and is not permutation invariant. Indeed, the direction of time affords crucial information about the dynamics of a task to the agent, e.g., a Half-Cheetah running forward versus backward has arguably the same state trajectory but in a different order. Further, the context in MQL is a deterministic function of the trajectory. Both these aspects are different than the context used by Rakelly et al. (2019) who design an inference network and sample a probabilistic context conditioned on a moving window. RL algorithms are quite complex and challenging to reproduce. Current meta-RL techniques which build upon them further exacerbate this complexity. Our demonstration that a simple context variable is enough is an important contribution.

### 3.2 ADAPTATION TO A NEW TASK

We next discuss the adaptation procedure which adapts the meta-trained policy $\widehat{\theta}_{\text{meta}}$ to a new task $D^{\text{new}}$ with few data. MQL optimizes the adaptation objective introduced in (2) into two steps.

**1. Vanilla off-policy adaptation:** The first step is to update the policy using the new data as

$$\arg\max_\theta \left\{ \mathop{\mathbb{E}}_{\tau \sim D^{\text{new}}} \left[ \ell^{\text{new}}(\theta) \right] - \frac{\lambda}{2} \|\theta - \widehat{\theta}_{\text{meta}}\|_2^2 \right\}. \tag{18}$$

The quadratic penalty $\|\theta - \widehat{\theta}_{\text{meta}}\|^2$ keeps the parameters close to $\widehat{\theta}_{\text{meta}}$. This is crucial to reducing the variance of the model that is adapted using few data from the new task (Reddi et al., 2015). Off-policy learning is critical in this step because of its sample efficiency. We initialize $\theta$ to $\widehat{\theta}_{\text{meta}}$ while solving (18).

**2. Importance-ratio corrected off-policy updates:** The second step of MQL exploits the meta-training replay buffer. Meta-training tasks $\mathcal{D}_{\text{meta}}$ are disjoint from $D^{\text{new}}$ but because they are expected to come from the same task distribution, transitions collected during meta-training can potentially be exploited to adapt the policy. This is difficult to do on two counts. First, the meta-training transitions do not come from $D^{\text{new}}$. Second, even for transitions from the same task, it is non-trivial to update the policy because of extrapolation error (Fujimoto et al., 2018a): the value function has high error on states it has not seen before. Our use of the propensity score to reweigh transitions is a simpler version of the conditional generative model used by Fujimoto et al. (2018a) in this context.

MQL fits a logistic classifier on a mini-batch of transitions from the meta-training replay buffer and the transitions collected from the new task in step 1. The context variable $z_t$ is the feature for this classifier. The logistic classifier estimates the importance ratio $\beta(\tau; D^{\text{new}}, \mathcal{D}_{\text{meta}})$ and can be used to reweigh data from the meta-training replay buffer for taking updates as

$$\arg\max_\theta \left\{ \mathop{\mathbb{E}}_{\tau \sim \mathcal{D}_{\text{meta}}} \left[ \beta(\tau; D^{\text{new}}, \mathcal{D}_{\text{meta}}) \, \ell^{\text{new}}(\theta) \right] - \frac{\lambda}{2} \|\theta - \widehat{\theta}_{\text{meta}}\|_2^2 \right\}. \tag{19}$$

We have again included a quadratic penalty $\|\theta - \widehat{\theta}_{\text{meta}}\|^2$ that keeps the new parameters close to $\widehat{\theta}_{\text{meta}}$. Estimating the importance ratio involves solving a convex optimization problem on few samples (typically, 200 from the new task and 200-400 from the meta-training tasks). This classifier allows MQL to exploit the large amount of past data. In practice, we perform as many as $100\times$ more weight updates using (19) than (18).

**Remark 3 (Picking the coefficient $\lambda$).** Following Fakoor et al. (2019), we pick

$$\lambda = 1 - \widehat{\text{ESS}}$$

for both the steps (18–19). This relaxes the quadratic penalty if the new task is similar to the meta-training tasks ($\widehat{\text{ESS}}$ is large) and vice-versa. While $\lambda$ could be tuned as a hyper-parameter, our empirical results show that adapting it using $\widehat{\text{ESS}}$ is a simple and effective heuristic.

**Remark 4 (Details of estimating the importance ratio).** It is crucial to ensure that the logistic classifier for estimating $\beta$ generalizes well if we are to reweigh transitions in the meta-training replay buffer that are different than the ones the logistic was fitted upon. We do so in two ways: (i) the regularization co-efficient in (11) is chosen to be relatively large, that way we prefer false negatives than risk false positives; (ii) transitions with very high $\beta$ are valuable for updating (19) but cause a large variance in stochastic gradient descent-based updates, we clip $\beta$ before taking the update in (19). The clipping constant is a hyper-parameter and is given in Sec. 4.

MQL requires having access to the meta-training replay buffer during adaptation. This is not a debilitating requirement and there are a number of clustering techniques that can pick important transitions from the replay-buffer if a robotic agent is limited by available hard-disk space. The meta-training replay buffer is at most 3 GB for the experiments in Sec. 4.

## 4 EXPERIMENTS

This section presents the experimental results of MQL. We first discuss the setup and provide details the benchmark in Sec. 4.1. This is followed by empirical results and ablation experiments in Sec. 4.2.

### 4.1 SETUP

**Tasks and algorithms:** We use the Mu-JoCo (Todorov et al., 2012) simulator with OpenAI Gym (Brockman et al., 2016) on continuous-control meta-RL benchmark tasks. These tasks have different rewards, randomized system parameters (Walker-2D-Params) and have been used in previous papers such as Finn et al. (2017); Rothfuss et al. (2018); Rakelly et al. (2019). We compare against standard baseline algorithms, namely MAML (TRPO (Schulman et al., 2015) variant) (Finn et al., 2017), RL2 (Duan et al., 2016), ProMP (Rothfuss et al., 2018) and PEARL (Rakelly et al., 2019). We obtained the training curves and hyper-parameters for all the three algorithms from the published code by Rakelly et al. (2019).

We will compare the above algorithms against: (i) vanilla TD3 (Fujimoto et al., 2018a) without any adaptation on new tasks, (ii) TD3-context: TD3 with GRU-based context Sec. 3.1.1 without any adaptation, and (iii) MQL: TD3 with context and adaptation on new task using the procedure in Sec. 3.2. All the three variants use

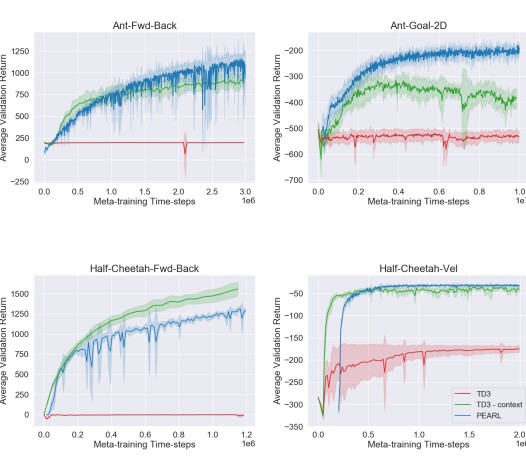

Figure 2: **Average undiscounted return of TD3 and TD3-context compared with PEARL for validation tasks from four meta-RL environments.** The agent fails to learn if the policy is conditioned only on the state. In contrast, everything else remaining same, if TD3 is provided access to context, the rewards are much higher. In spite of not adapting on the validation tasks, TD3-context is comparable to PEARL.

the multi-task objective for meta-training (15).
We use Adam (Kingma & Ba, 2014) for opti-
mizing all the loss functions in this paper.

**Evaluation:** Current meta-RL benchmarks lack a systematic evaluation procedure. [2] For each environment, Rakelly et al. (2019) constructed a fixed set of meta-training tasks ($\mathcal{D}_{\text{meta}}$) and a validation set of tasks $D^{\text{new}}$ that are disjoint from the meta-training set. To enable direct comparison with published empirical results, we closely followed the evaluation code of Rakelly et al. (2019) to create these tasks. We also use the exact same evaluation protocol as that of these authors, e.g., 200 time-steps of data from the new task, or the number of evaluation episodes. We report the undiscounted return on the validation tasks with statistics computed across 5 random seeds.

## 4.2    RESULTS

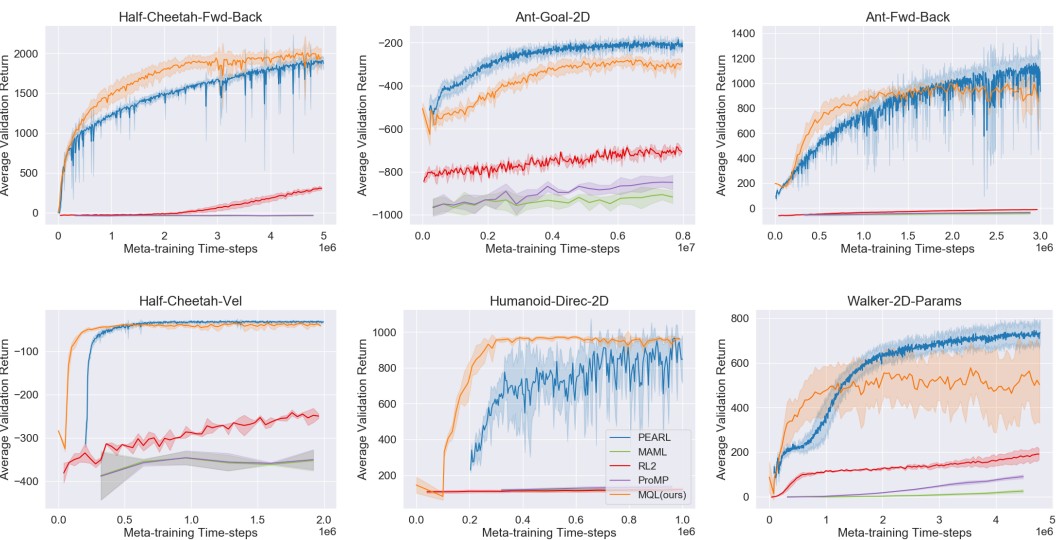

Figure 3: **Comparison of the average undiscounted return of MQL (orange) against existing meta-RL algorithms on continuous-control environments.** We compare against four existing algorithms, namely MAML (green), RL2 (red), PROMP (purple) and PEARL (blue). In all environments except Walker-2D-Params and Ant-Goal-2D, MQL is better or comparable to existing algorithms in terms of both sample complexity and final returns.

Our first result, in Fig. 2, is to show that **vanilla off-policy learning with context, without any adaptation is competitive with state of the art** meta-RL algorithms. We used a standard implementation of TD3 and train on the meta-training tasks using the multi-task objective (15). Hyper-parameters for these tasks are provided in Appendix D. This result is surprising and had gone unnoticed in the current literature. Policies that have access to the context can easily generalize to the validation tasks and achieve performance that is comparable to more sophisticated meta-RL algorithms.

We next evaluate MQL against existing meta-RL benchmarks on all environments. The results are shown in Fig. 3. We see that for all environments except Walker-2D-Params and Ant-Goal-2D, MQL obtains comparable or better returns on the validation tasks. In most cases, in particular for the challenging Humanoid-Direc-2D environment, MQL converges faster than existing algorithms. MAML and ProMP require about 100M time-steps to converge to returns that are significantly worse

---

[2]For instance, training and validation tasks are not explicitly disjoint in Finn et al. (2017); Rothfuss et al. (2018) and these algorithms may benefit during adaptation from having seen the same task before. The OpenAI Gym environments used in Finn et al. (2017); Rothfuss et al. (2018); Rakelly et al. (2019) provide different rewards for the same task. The evaluation protocol in existing papers, e.g., length of episode for a new task, amount of data available for adaptation from the new task, is not consistent. This makes reproducing experiments and comparing numerical results extremely difficult.

than the returns of off-policy algorithms like MQL and PEARL. Compare the training curve for TD3-context for the Ant-Goal-2D environment in Fig. 2 with that of the same environment in Fig. 3: the former shows a prominent dip in performance as meta-training progresses; this dip is absent in Fig. 3 and can be attributed to the adaptation phase of MQL.

## 4.3 ABLATION EXPERIMENTS

We conduct a series of ablation studies to analyze the different components of the MQL algorithm. We use two environments for this purpose, namely Half-Cheetah-Fwd-Back and Ant-Fwd-Back. Fig. 4a shows that the adaptation in MQL in (18) and (19) improves performance. Also observe that MQL has a smaller standard deviation in the returns as compared to TD3-context which does not perform any adaptation; this can be seen as the adaptation phase making up for the lost performance of the meta-trained policy on a difficult task. Next, we evaluate the importance of the additional data from the replay buffer in MQL. Fig. 4b compares the performance of MQL with and without updates in (19). We see that the old data, even if it comes from different tasks, is useful to improve the performance on top of (18). Fig. 4c shows the effectiveness of setting $\lambda = 1 - \widehat{\text{ESS}}$ as compared to a fixed value of $\lambda = 0.5$. We see that modulating the quadratic penalty with $\widehat{\text{ESS}}$ helps, the effect is minor for Sec. 4.3. The ideal value of $\lambda$ depends on a given task and using $1 - \widehat{\text{ESS}}$ can help to adjust to different tasks without the need to do hyper-parameter search per task. Finally, Fig. 5 shows the evolution of $\lambda$ and $\beta(z)$ during meta-training. The coefficient $\lambda$ is about 0.55 and $\beta(z)$ is 0.8 for a large fraction of the time. The latter indicates that propensity score estimation is successful in sampling transitions from the meta-training replay buffer that are similar to the validation tasks. The value of $\lambda$ remains relatively unchanged during training. This value indicates the fraction of transitions in the old data that are similar to those from the new tasks; since there are two distinct tasks in Ant-Fwd-Back, the value $\lambda = 0.55$ is appropriate.

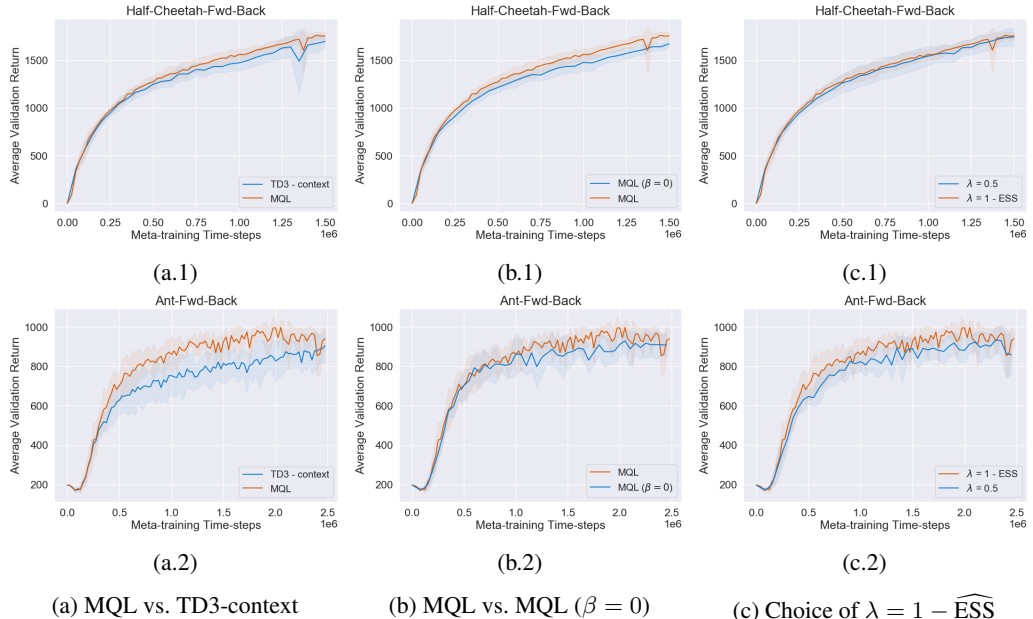

| (a.1) | (b.1) | (c.1) |
| :---: | :---: | :---: |

| (a.2) | (b.2) | (c.2) |
| :---: | :---: | :---: |

| (a) MQL vs. TD3-context | (b) MQL vs. MQL ($\beta = 0$) | (c) Choice of $\lambda = 1 - \widehat{\text{ESS}}$ |
| :---: | :---: | :---: |

Figure 4: **Ablation studies to examine various components of MQL.**

## 4.4 RELATED WORK

**Learning to learn:** The idea of building an inductive bias for learning a new task by training on a large number of related tasks was established in a series of works (Utgoff, 1986; Schmidhuber, 1987; Baxter, 1995; Thrun, 1996; Thrun & Pratt, 2012). These papers propose building a base learner that fits on each task and a meta-learner that learns properties of the base learners to output a new base

learner for a new task. The recent literature instantiates this idea in two forms: (i) the meta-learner directly predicts the base-learner (Wang et al., 2016; Snell et al., 2017) and (ii) the meta-learner learns the updates of the base-learner (Bengio et al., 1992; Hochreiter et al., 2001; Finn et al., 2017).

**Meta-training versus multi-task training:** Meta-training aims to train a policy that can be adapted efficiently on a new task. Conceptually, the improved efficiency of a meta-learner comes from two things: (i) building a better inductive bias to initialize the learning (Schmidhuber et al., 1997; Baxter, 1995; 2000; Mitchell, 1980), or (ii) learning a better learning procedure (Bengio et al., 1997; Lee et al., 2019). The two notions of meta-learning above are complementary to each other and in fact, most recent literature using deep neural networks, e.g., MAML (Finn et al., 2017) and Prototypical Networks (Snell et al., 2017) confirms to the first notion of building a better inductive bias.

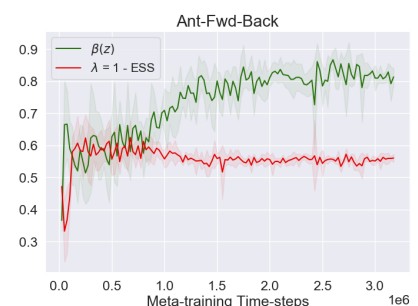

Figure 5: **Evolution of $\lambda$ and $\beta(z)$ during meta-training.**

The multi-task training objective in MQL is the simplest possible instantiation of this idea: it maximizes the average reward on all tasks and learns a better prior without explicitly training for improving adaptation. This aspect of MQL coincides with a recent trend in meta-learning for image classification where it has been observed that modifications to episodic meta-training (Snell et al., 2017; Gidaris & Komodakis, 2018; Chen et al., 2018), or even foregoing meta-training completely (Dhillon et al., 2019) performs better. We speculate two reasons for this phenomenon: (i) meta-training methods are complex to implement and tune, and (ii) powerful function classes such as deep neural networks may have leftover capacity to adapt to a new task even if they are not explicitly trained for adaptation.

**Context-based approaches:** Both forms of meta-learning above have been employed relatively successfully for image classification (Snell et al., 2017; Ravi & Larochelle, 2016; Finn et al., 2017). It has however been difficult to replicate that empirical performance in RL: sensitivity to hyper-parameters (Henderson et al., 2018) precludes directly predicting the base-learner while long-range temporal dependencies make it difficult to learn the updates of the base learner (Nichol et al., 2018). Recent methods for meta-RL instead leverage context and learn a policy that depends on just on the current state $x_t$ but on the previous history. This may be done in a recurrent fashion (Heess et al., 2015; Hausknecht & Stone, 2015) or by learning a latent representation of the task (Rakelly et al., 2019). Context is a powerful construct: as Fig. 1 shows, even a simple vanilla RL algorithm (TD3) when combined with context performs comparably to state-of-the-art meta-RL algorithms. However, context is a meta-training technique, it does not suggest a way to adapt a policy to a new task. For instance, Rakelly et al. (2019) do not update parameters of the policy on a new task. They rely on the latent representation of the context variable generalizing to new tasks. This is difficult if the new task is different from the training tasks; we discuss this further in Sec. 3.1.1.

**Policy-gradient-based algorithms versus off-policy methods:** Policy-gradient-based methods have high sample complexity (Ilyas et al., 2018). This is particularly limiting for meta-RL (Finn et al., 2017; Rothfuss et al., 2018; Houthooft et al., 2018) where one (i) trains on a large number of tasks and, (ii) aims to adapt to a new task with few data. Off-policy methods offer substantial gains in sample complexity. This motivates our use of off-policy updates for both meta-training and adaptation. Off-policy updates allow using past data from other policies. MQL exploits this substantially, it takes up to $100\times$ more updates using old data than new data during adaptation. Off-policy algorithms are typically very sensitive to hyper-parameters (Fujimoto et al., 2018a) but we show that MQL is robust to such sensitivity because it adapts automatically to the distribution shift using the Effective Sample Size (ESS).

**Propensity score estimation** has been extensively studied in both statistics (Robert & Casella, 2013; Quionero-Candela et al., 2009) and RL (Dudík et al., 2011; Jiang & Li, 2015; Kang et al., 2007; Bang & Robins, 2005). It is typically used to reweigh data from the proposal distribution to compute estimators on the target distribution. MQL uses propensity scores in a novel way: we fit a propensity score estimator on a subset of the meta-training replay buffer and use this model to sample transitions

from the replay buffer that are *similar* to the new task. The off-policy updates in MQL are essential to exploiting this data. The coefficient of the proximal term in the adaptation-phase objective (18–19) using the effective sample size (ESS) is inspired from the recent work of Fakoor et al. (2019).

## 5  DISCUSSION

The algorithm proposed in this paper, namely MQL, builds upon on three simple ideas. First, Q-learning with context is sufficient to be competitive on current meta-RL benchmarks. Second, maximizing the average reward of training tasks is an effective meta-learning technique. The meta-training phase of MQL is significantly simpler than that of existing algorithms and yet it achieves comparable performance to the state of the art. This suggests that we need to re-think meta-learning in the context of rich function approximators such as deep networks. Third, if one is to adapt to new tasks with few data, it is essential to exploit every available avenue. MQL recycles data from the meta-training replay buffer using propensity estimation techniques. This data is essentially free and is completely neglected by other algorithms. This idea can potentially be used in problems outside RL such as few-shot and zero-shot image classification.

Finally, this paper sheds light on the nature of benchmark environments in meta-RL. The fact that even vanilla Q-learning with a context variable—without meta-training and without any adaptation—is competitive with state of the art algorithms indicates that (i) training and validation tasks in the current meta-RL benchmarks are quite similar to each other and (ii) current benchmarks may be insufficient to evaluate meta-RL algorithms. Both of these are a call to action and point to the need to invest resources towards creating better benchmark problems for meta-RL that drive the innovation of new algorithms.

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

# A   PSEUDO-CODE

The pseudo-code for MQL during training and adaption are given in Algorithm 1 and Algorithm 2. After MQL is trained for a given environment as described in Algorithm 1, it returns the meta-trained policy $\theta$ and replay buffer containing train tasks.

Next, Algorithm 2 runs the adaptation procedure which adapts the meta-trained policy to a test task $D$ with few data. To do so, MQL optimizes the adaptation objective into two steps. After gathering data from a test task $D$, MQL first updates the policy using the new data (line 4). MQL then fits a logistic classifier on a mini-batch of transitions from the meta-training replay buffer and the transitions collected from the test task and then estimates $\widehat{\text{ESS}}$ (lines 5-6). Finally, the adaptation step runs for $n$ iterations (lines 7 - 10) in which MQL can exploit past data in which it uses propensity score to decide whether or not a given sample is related to the current test task.

---

**Algorithm 1:** MQL - Meta-training

   **Input:** Set of training tasks $\mathcal{D}_{\text{meta}}$

1  Initialize the replay buffer
2  Initialize parameters $\theta$ of an off-policy method, e.g., TD3
3  **while** *not done* **do**
4     // Rollout and update policy
5     Sample a task $D \sim \mathcal{D}_{\text{meta}}$
6     Gather data from task $D$ using policy $\pi_\theta$ while feeding transitions through context GRU. Add trajectory to the replay buffer.
7     $b \leftarrow$ Sample mini-batch from buffer
8     Update parameters $\theta$ using mini-batch $b$ and Eqn. (15)
9  $\theta_{\text{meta}} \leftarrow \theta$
10  **return** $\theta_{\text{meta}}$ , replay buffer

---

**Algorithm 2:** MQL - Adaptation

   **Input:** Test task $D$, meta-training replay buffer, meta-trained policy $\theta_{\text{meta}}$

1  Initialize temporary buffer buf
2  $\theta \leftarrow \theta_{meta}$
3  buf $\leftarrow$ Gather data from $D$ using $\pi_{\theta_{meta}}$
4  Update Eqn. (18) using buf
5  Fit $\beta(D)$ using buf and meta-training replay buffer using Eqn. (12)
6  Estimate $\widehat{\text{ESS}}$ using $\beta(D)$ using Eqn. (13)
7  **for** $i \le n$ **do**
8     $b \leftarrow$ sample mini-batch from meta-training replay buffer
9     Calculate $\beta$ for $b$
10     Update $\theta$ using Eqn. (19)
11  Evaluate $\theta$ on a new rollout from task $D$
12  **return** $\theta$

---

# B   OUT-OF-DISTRIBUTION TASKS

MQL is designed for explicitly using data from the new task along with off-policy data from old, possibly very different tasks. This is on account of two things: (i) the loss function of MQL does not use the old data if it is very different from the new task, $\beta$ is close to zero for all samples, and (ii) the first term in (18) makes multiple updates using data from the new task. To explore this aspect, we create an out-of-distribution task using the "Half-Cheetah-Vel" environment wherein we use disjoint sets of velocities for meta-training and testing. The setup is as follows:

- **Half-Cheetah-Vel-OOD-Medium:** target velocity for a training task is sampled uniformly randomly from $[0, 2.5]$ while that for test task is sampled uniformly randomly from $[2.5, 3.0]$.

This is what we call "medium" hardness task because although the distributions of train and test velocities is disjoint, they are close to each other.

- **Half-Cheetah-Vel-OOD-Hard:** target velocity for a training task is sampled uniformly randomly from $[0, 1.5]$ while that for test task is sampled uniformly randomly from $[2.5, 3.0]$. This is a "hard" task because the distributions of train and test velocities are far away from each other.

Fig. 6a shows that MQL significantly outperforms PEARL when the train and test target velocities come from disjoint sets. We used the published code of PEARL (Rakelly et al., 2019) for this experiment. This shows that the adaptation in MQL is crucial to generalizing to new situations which are not a part of the meta-training process. Fig. 6b shows the evolution of the proximal penalty coefficient $\lambda$ and the propensity score $\beta(z)$ during meta-training for the medium-hard task. We see that $\lambda \approx 0.8$ while $\beta(z) \approx 0.2$ throughout training. This indicates that MQL automatically adjusts its test-time adaptation to use only few samples in (19) if the test task provides transitions quite different than those in the replay buffer.

We next discuss results on the harder task **Half-Cheetah-Vel-OOD-Hard**. There is a very large gap between training and test target velocities in this case. Fig. 7a shows the comparison with the same test protocol as the other experiments in this paper. In particular, we collect 200 time-steps from the new task and use it for adaptation in both MQL and TD3-context. Since this task is particularly hard, we also ran an experiment where 1200 time-steps (6 episodes) are given to the two algorithms for adaptation. The results are shown in Fig. 7b. In both cases, we see that MQL is better than TD3-context by a large margin (the standard deviation on these plots is high because the environment is hard). Note that since we re-initialize the hidden state of the context network at the beginning of each episode, TD3-context cannot take advantage of the extra time-steps. MQL on the other hand updates the policy explicitly and can take advantage of this extra data.

For sake of being thorough, we collected 800 time-steps from the new task from the same episode, the results are shown in Fig. 8a. We again notice that MQL results in slightly higher rewards than TD3-context in spite of the fact that both the algorithms suffer large degradation in performance as compared to Figs. 7a and 7b.

Figs. 7c, 7d and 8b show that the proximal penalty coefficient $\lambda \approx 1$ and the propensity score $\beta(z) \approx 0$ for a large fraction of training. This proves that MQL is able to automatically discard samples unrelated to the new test during the adaptation phase.

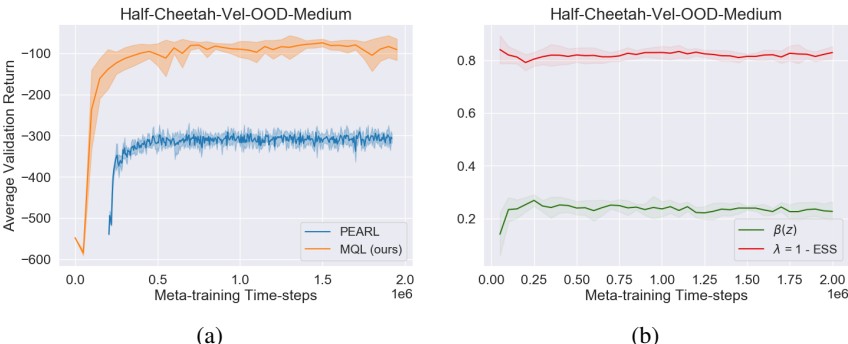

(a)                                            (b)

Figure 6: **Comparison of the average return of MQL (orange) against existing PEARL algorithms (blue).** Fig. 6a shows that MQL significantly outperforms PEARL when the train and test target velocities come from disjoint sets. Fig. 6b shows the evolution of the proximal penalty coefficient $\lambda$ and the propensity score $\beta(z)$. We see that $\beta(z)$ is always small which demonstrates that MQL automatically adjusts the adaptation in (19) if the test task is different from the training tasks.

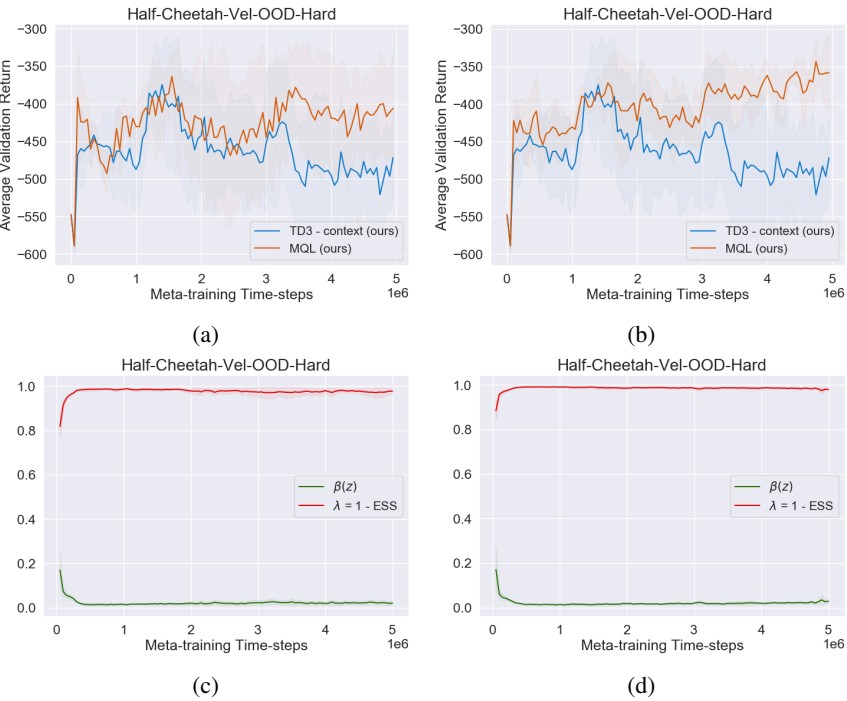

Figure 7: (a,b) **Comparison of the average return of MQL (orange) against TD3-context (blue).** Fig. 7a shows the comparison with the same test protocol as the other experiments in this paper. In particular, we collect 200 time-steps from the new task and use it for adaptation in both MQL and TD3-context. Since this task is particularly hard, we also ran an experiment where 1200 time-steps (6 episodes) where results are shown in Fig. 7b. In both cases, we see that MQL is better than TD3-context by a large margin (the standard deviation on these plots is high because the environment is hard). (c,d) **Evolution of $\lambda$ and $\beta(z)$ during meta-training.** shows the evolution of the proximal penalty coefficient $\lambda$ and the propensity score $\beta(z)$. We see in Fig. 7c and Fig. 7d that $\beta(z)$ is always small which demonstrates that MQL automatically adjusts the adaptation if the test task is different from the training tasks.

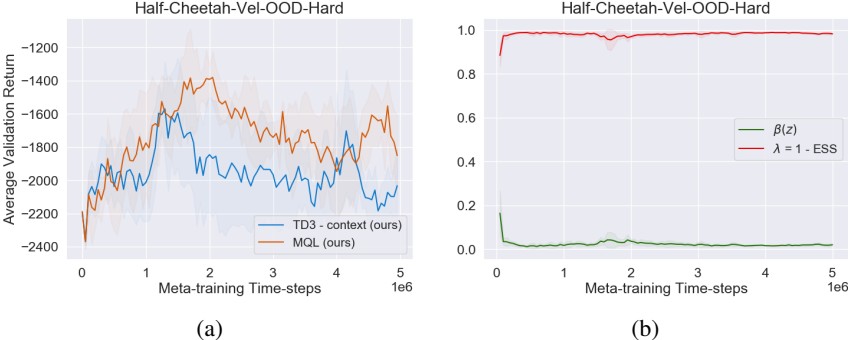

Figure 8: (a) **Comparison of the average return of MQL (orange) against TD3-context (blue).** For these experiments, we collected 800 time-steps from the new task from the same episode, the results are shown in Fig. 8a. We again notice that MQL results in slightly higher rewards than TD3-context in spite of the fact that both the algorithms suffer large degradation in performance as compared to Figs. 7a and 7b. (b) **Evolution of $\lambda$ and $\beta(z)$ during meta-training** shows the evolution of the proximal penalty coefficient $\lambda$ and the propensity score $\beta(z)$.

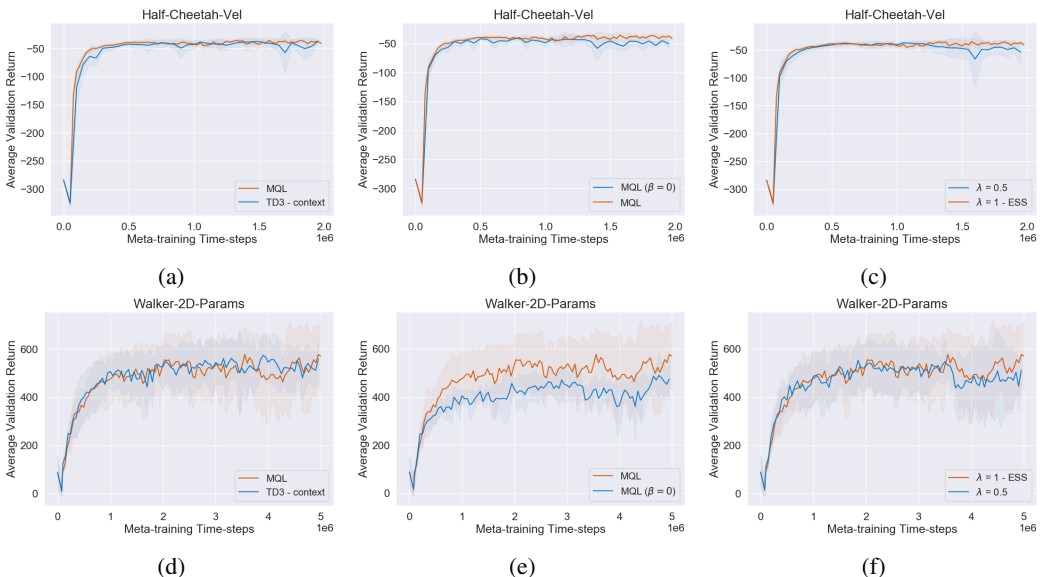

Figure 9: **Ablation studies to examine various components of MQL.** Fig. 9a shows that the adaptation in MQL in (18) and (19) improves performance. One reason for that is because test and training tasks in Walker-2D-Params are very similar as it shown in Fig. 10b. Next, we evaluate the importance of the additional data from the replay buffer in MQL. Fig. 9b compares the performance of MQL with and without updates in (19). We see that the old data, even if it comes from different tasks, is useful to improve the performance on top of (18). Fig. 9c and Fig. 9f show the effectiveness of setting $\lambda = 1 - \widehat{\text{ESS}}$ as compared to a fixed value of $\lambda = 0.5$. We see that modulating the quadratic penalty with $\widehat{\text{ESS}}$ helps, the effect is minor for Sec. 4.3. The ideal value of $\lambda$ depends on a given task and using $1 - \widehat{\text{ESS}}$ can help to adjust to different tasks without the need to do hyper-parameter search per task.

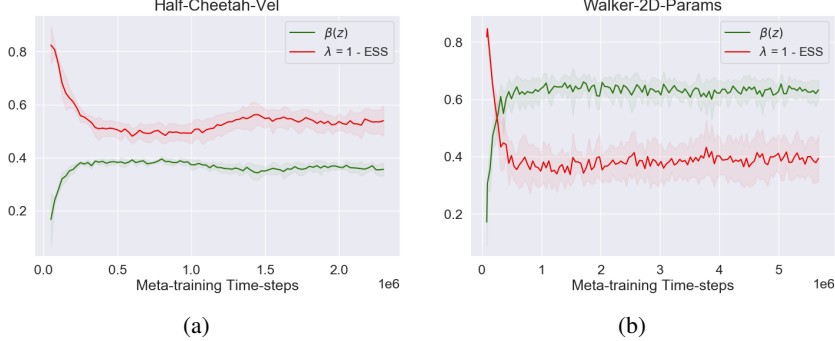

Figure 10: **Evolution of $\lambda$ and $\beta(z)$ during meta-training** shows the evolution of the proximal penalty coefficient $\lambda$ and the propensity score $\beta(z)$. We see in Fig. 10a that $\beta(z)$ stays around 0.4 which demonstrates MQL automatically adjusts the adaptation if the test task is different from the training tasks. Fig. 10b shows that test and training tasks are very similar as $\beta(z)$ is around 0.6.

## C    MORE ABLATION STUDIES

We conduct a series of additional ablation studies to analyze the different components of the MQL algorithm. We use two environments for this purpose, namely Half-Cheetah-Vel and Walker-2D-Params. Fig. 9 and Fig. 10 show the result of these experiments. These experiments show that adaptation phase is more useful for Half-Cheetah-Vel than Walker-2D-Params as test and training tasks are very similar in Walker-2D-Params which helps TD3-context achieves strong performance that leaves no window for improvement with adaptation.

## D    HYPER-PARAMETERS AND MORE DETAILS OF THE EMPIRICAL RESULTS

Table 1: **Hyper-parameters for MQL and TD3 for continuous-control meta-RL benchmark tasks.** We use a network with two full-connected layers for all environments. The batch-size in Adam is fixed to 256 for all environments. The abbreviation HC stands for Half-Cheetah. These hyper-parameters were tuned by grid-search.

|  | Humanoid | HC-Vel | Ant-FB | Ant-Goal | Walker | HC-FB |
|---|---|---|---|---|---|---|
| $\beta$ clipping | 1 | 1.1 | 1 | 1.2 | 2 | 0.8 |
| TD3 exploration noise | 0.3 | 0.3 | 0.3 | 0.2 | 0.3 | 0.2 |
| TD3 policy noise | 0.2 | 0.3 | 0.3 | 0.4 | 0.3 | 0.2 |
| TD3 policy update frequency | 2 | 2 | 2 | 4 | 4 | 3 |
| Parameter updates per iteration (meta-training) | 500 | 1000 | 200 | 1000 | 200 | 200 |
| Adaptation parameter updates per episode (eq. 18) | 10 | 5 | 10 | 5 | 10 | 10 |
| Adaptation parameter updates per episode (eq. 19) | 200 | 400 | 100 | 400 | 400 | 300 |
| GRU sequence length | 20 | 20 | 20 | 25 | 10 | 10 |
| Context dimension | 20 | 20 | 15 | 30 | 30 | 30 |
| Adam learning rate | 0.0003 | 0.001 | 0.0003 | 0.0004 | 0.0008 | 0.0003 |

