# OpenReview forum: "Meta-Q-Learning"
_ICLR.cc/2020/Conference — Accept (Talk)_

### Official Review · AnonReviewer1 · 2019-10-20
**Official Blind Review #1**

**Rating:** 6

**Review:**

The authors investigate meta-learning in reinforcement learning with respect to sample efficiency and the necessity of meta-learning an adaptation scheme. Based on their findings, they propose a new algorithm 'MQL' (Meta-Q-Learning) that is off-policy and has a fixed adaptation scheme but is still competitive on meta-RL benchmarks (a distribution of environments that differ slightly in their reward functions).

They motivate the paper by data-inefficiency of current meta-learning approaches and empirical results suggesting that meta-learning the adaptation scheme is less important than feature reuse.

On the other hand, their introduction section would benefit from additional references to the kind of meta-learning they describe. In particular, their so-called "definition of meta-learning" is mostly about domain randomization (e.g. Tobin et al. 2017 https://arxiv.org/abs/1703.06907) and not about the broader 'learning to learn' RL methodology (in particular Schmidhuber 1994 "On learning how to learn learning strategies").

**The authors make the following contributions:**

1. They show that Q-Learning trained on multiple tasks with a context variable as an input (an RNN state summarizing previous transitions) is competitive to related work when evaluated on a test task even though no adaptation is performed

2. Based on these observations, they introduce a new method for off-policy RL that does not directly optimize for adaptation but instead uses a fixed adaptation scheme

3. The new method leverages data during meta-testing that was collected during meta-training using importance weights for increased sample efficiency

**Overall, we believe the contributions are significant and sufficiently empirically justified.**

There are strong similarities, however, to parallel work on analyzing whether MAML relies on feature reuse or rapid learning (Raghu et al. 2019 https://arxiv.org/abs/1909.09157).

This work and the present submission conclude that feature reuse is much more significant than meta-learning an adaptation scheme when evaluated on current meta-RL benchmarks. This is a significant result and supports the new method developed in this paper.

During meta-training, their proposed method maximizes only the average return across tasks, not the ability to adapt from the resulting parameters.

Their method introduces a fixed (non-learned) adaptation scheme that performs favorably compared to certain methods from the existing meta-learning literature and demonstrates that even dropping this adaptation still does well.

There are strong similarities to Nichol et al. 2018 (https://arxiv.org/abs/1803.02999). We encourage the authors to relate this work to Raghu et al. 2019 and Nichol et al. 2018.

**Despite these interesting results, we strongly disagree with the meta-learning narrative of their new method.**

Because the adaptation scheme is no longer optimized directly, instead a fixed adaptation scheme is assumed, hence the approach in this paper is no longer a meta-learning algorithm.

Instead, this method has strong similarities with transfer-learning and domain adaptation (first training on one distribution of tasks, then fine-tuning on another task).

The authors should discuss the links to these fields of research, and clarify what's really novel, already in the abstract.

For example, on page 2 the authors claim that optimizing the multi-task objective (the mean error across tasks) is the simplest form of meta-learning. This objective, however, is NOT meta-learning.

**Decision.**

The submission contains strong empirical results emphasizing the significance of feature reuse and the insignificance of learned adaptation on the tested meta-RL benchmarks.

The reuse of experience from meta-training during meta-testing by employing importance weights is also an interesting contribution.

In contrast, we are not satisfied with the presentation of their new approach as a meta-learning approach. This method should be introduced along the lines of: 'Transfer Learning / Feature Reuse in RL is competitive to meta-learning across similar tasks'.

In its current form, we tend to reject the paper because it further obscures what the term meta-learning refers to. The authors are confusing it with more limited transfer learning.

Additionally, it was not clear to us whether the quadratic penalty they add to their adaptation scheme is only empirically valid or whether there is a theoretical reason.

For now, we'd lean towards rejecting this submission, but we might change our minds, provided the comments above were addressed in a satisfactory way - let us wait for the rebuttal.

Edit after rebuttal: score increased!

**Experience Assessment:**

I have published in this field for several years.

**Review Assessment: Checking Correctness Of Derivations And Theory:**

I assessed the sensibility of the derivations and theory.

**Review Assessment: Checking Correctness Of Experiments:**

I assessed the sensibility of the experiments.

**Review Assessment: Thoroughness In Paper Reading:**

I read the paper thoroughly.

---

> ### Author Response · Authors · 2019-11-14
> **Response to Review #1 (Part 1)**
>
> Thank you for your feedback. Please also see the main comment above. The reviewer says, “the contributions of this paper are significant and sufficiently empirically justified”, “the submission contains strong empirical results emphasizing the significance of feature reuse and the insignificance of learned adaptation on the tested meta-RL benchmarks.”, “This is a significant result and supports the new method developed in this paper.”. We are extremely puzzled at the low score. We hope you will consider increasing your score after seeing our response.
>
>
> >> “definition of meta-learning" is mostly about domain randomization”, “we are not satisfied with the presentation of their new approach as a meta-learning approach. This method should be introduced along the lines of: 'Transfer Learning / Feature Reuse in RL is competitive to meta-learning across similar tasks”, “Despite these interesting results, we strongly disagree with the meta-learning narrative of their new method”, “we tend to reject the paper because it further obscures what the term meta-learning refers to. The authors are confusing it with more limited transfer learning.”
>
> A meta-learner is an algorithm that has improved efficiency (statistically or computationally) of learning a new task by virtue of having seen related tasks in the past. This is the definition in "On learning how to learn learning strategies" by Schmidhuber 1994, “Is learning the n^th thing any easier than learning the first?” by Thrun 1996 , “Learning to learn” by Thrun & Pratt 1996, etc. The improved efficiency of a meta-learner can be the result of two things: (i) a better prior (“shift its inductive bias” as Schmidhuber 1994 notes), or (ii) a better learning procedure that starts from the same prior, e.g., “On the optimization of a synaptic learning rule” by Bengio et al., 1992 or “Meta-Learning with Differentiable Convex Optimization” by Lee et al., 2019. The two notions of meta-learning above are complementary to each other and in fact, most recent literature using deep neural networks, e.g., MAML by Finn et al., 2017, Prototypical Networks by Snell et al., 2017 etc. confirms to the first notion of building a better prior.
>
> Our definition of meta-learning is (i). The reviewer notes that “adaptation scheme is no longer optimized directly”. And this is one key point of our paper. We perform multi-task learning at training time to learn a better prior for adaptation. More the number of training tasks better the prior and more sample efficient the adaptation.
>
> The current literature lacks a mathematically rigorous definition of meta-learning and we take the reviewer’s concerns in the right spirit. We have added the above comments to the related work in Section 4.4.
>
> We are at a loss at the connection of other concepts in machine learning with meta-learning by the reviewer. Domain randomization is a data augmentation technique, it is not meta-learning. Transfer Learning only adapts from one source task to another target task, it does not start from a family of training tasks; it is not meta-learning. We do not perform feature reuse in MQL: we update the entire policy using both data from the new task and the meta-training tasks.
>
> >> During meta-training, their proposed method maximizes only the average return across tasks, not the ability to adapt from the resulting parameters. For example, on page 2 the authors claim that optimizing the multi-task objective (the mean error across tasks) is the simplest form of meta-learning. This objective, however, is NOT meta-learning.
> This is a key point of our paper. Indeed, we perform multi-task learning at training time. We do not train for the ability to adapt the parameters. We instead learn a better prior for adaptation. The multi-task objective is the simplest possible instantiation of meta-training, as suggested by the notion (i) in the previous remark because it provides a better prior than initializing the weights randomly before adaptation. More the number of tasks during training time better the multi-task learnt prior. The procedure is also similar to an existing state-of-the-art algorithm meta-learning named PEARL (“Efficient Off-Policy Meta-Reinforcement Learning via Probabilistic Context Variables” by Rakelly et al., 2019) which trains a value function conditioned on a probabilistic context variable on all the training tasks.

---

> > ### Author Response · Authors · 2019-11-14
> > **Response to Review #1 (Part 2)**
> >
> >
> > >> There are strong similarities, however, to parallel work on analyzing whether MAML relies on feature reuse or rapid learning (Raghu et al. 2019 https://arxiv.org/abs/1909.09157)
> > This paper appeared on Arxiv less than a week before our ICLR submission. We however do not see any similarities with our work. Raghu et al. study feature reuse in meta-learning, we explicitly adapt the meta-learnt policy using old and new data. MQL is specialized to off-policy RL: it explicitly uses past data from related tasks to improve sample efficiency of adaptation. MQL adapts the entire policy, updates are not limited to only a few layers as in Raghu et al.. Further, the results of MQL on RL tasks are significantly better than those in Raghu et al.
> >
> > >> There are strong similarities to Nichol et al. 2018
> > The _only_ overlap is equation (16) which is taken from Nichol et al., 2018, with appropriate citation. We do not claim this to be new, or a contribution of our paper. We simply make an observation using (16) that the critical points of the multi-task loss are the same as those of MAML which motivates the meta-training procedure in MQL.
> >
> > >> it was not clear to us whether the quadratic penalty they add to their adaptation scheme is only empirically valid or whether there is a theoretical reason
> > The quadratic penalty keeps the policy being adapted close to the meta-trained policy. The theoretical reason for doing so is that the variance of adapted model can be very high with access to only limited data from the new task. The proximal term reduces this variance. See “Doubly Robust Covariate Shift Correction” by Reddi et al., 2015 for a proof. We have added a sentence to this effect in Section 3.2.

---

### Official Review · AnonReviewer3 · 2019-10-22
**Official Blind Review #3**

**Rating:** 8

**Review:**

Summary
-------------
The authors propose meta Q-learning, an algorithm for off-policy meta RL. The idea is to meta-train a context-dependent policy to maximize the expected return averaged over all training tasks, and then adapt this policy to any new task by leveraging both novel and past experience using importance sampling corrections. The proposed approach is evaluated on standard Mujoco benchmarks and compared to other relevant meta-rl algorithms.

Comments
--------------

Meta-rl is a relevant direction for mitigating the sample-complexity of rl agents and allowing them to scale to larger domains. This work proposes interesting ideas and overall it constitutes an nice contribution. In particular, I found interesting (and at the same time worrying) that a simple q-learning algorithm with hidden contexts compares favorably to state-of-art meta-rl approaches in standard benchmarks. The paper is well-organized and easy to read. Some comments/questions follow.

1. (15) is probably miss-written (theta is trained to maximize the TD error)

2. In the adaptation phase, are (18) and (19) performed one after the other? Could they be done at the same time by setting to 1 the importance weights of the new trajectories and sampling from the whole experience (new and old)?

3. Note that the ESS estimator (13) diverges to infinity when all weights are close to zero. Is it clipped to [0,1] in the experiments? See e.g. [1] or [2] for more robust estimators that are bounded.

4. Since many recent works try to improve the generalization capabilities of meta-rl algorithms, I was wondering how the proposed approach generalizes to out-of-distribution tasks (i.e., tasks that are unlikely to occur at meta-training). Though it is never mentioned in the paper, I believe the proposed method has the potential to be robust to negative transfer since the importance weights (which would be very small for very different tasks) should automatically discard old data and focus on new data alone. This is in contrast to many existing methods where the meta-trained model might negatively bias the learning of very different tasks. I think an experiment of this kind would be valuable to improve the paper.

[1] Elvira, V., Martino, L., & Robert, C. P. (2018). Rethinking the effective sample size. arXiv preprint arXiv:1809.04129.
[2] Tirinzoni, A., Salvini, M., & Restelli, M. (2019, May). Transfer of Samples in Policy Search via Multiple Importance Sampling. In International Conference on Machine Learning (pp. 6264-6274).

**Experience Assessment:**

I have published one or two papers in this area.

**Review Assessment: Checking Correctness Of Derivations And Theory:**

I assessed the sensibility of the derivations and theory.

**Review Assessment: Checking Correctness Of Experiments:**

I assessed the sensibility of the experiments.

**Review Assessment: Thoroughness In Paper Reading:**

I read the paper at least twice and used my best judgement in assessing the paper.

---

> ### Author Response · Authors · 2019-11-14
> **Response to Review #3**
>
> Thank you for your feedback and suggested experiments. Please also see the main comment above. We hope you will consider increasing your score after seeing our response.
>
> >> (15) is probably miss-written
> Yes. We have fixed this.
>
> >> (18) and (19) performed one after the other? Could they be done at the same time?
> Yes, (18) and (19) can be performed at the same time. We chose to implement them in two steps for ease of implementation. Please also see the pseudo-code in Appendix A.
>
> >> ESS estimator (13) diverges to infinity
> No. ESS does not diverge to infinity, it is upper bounded by the number of data and lower bounded by 1. This is seen by noting that ESS = N/(1 + var(w); we simply normalize ESS by the number of data to get hat(ESS). This is also the subject of Section 3.3 in Elvia et al., 2018 “Rethinking the effective sample size”.
>
> >> Out-of-distribution tasks
> Thanks, this is a great suggestion. We did an experiment on the Half-Cheetah-Vel environment by making the set of meta-training and validation target velocities disjoint. We find that the adaptation of MQL is essential to improve performance in this case. In contrast, both our TD3-context and PEARL perform poorly on this out-of-distribution task. Please see the common comment above for the results.
>
> >> MQL is in contrast to many existing methods where the meta-trained model might negatively bias the learning of very different tasks
> This is indeed a benefit of our method: MQL does not suffer from negative transfer. If the new task is very different from the meta-training tasks then beta is close to zero which automatically discards old data. The proximal penalty is large which prevents the policy from changing too much under such a distribution shift. But the first term in (18) still benefits from new data.

---

> > ### Comment · AnonReviewer3 · 2019-11-15
> > **Post-rebuttal comments**
> >
> > Dear authors,
> >
> > Thank you for your response and for running the additional experiments. As mentioned in the review, I was already convinced about the contributions of this work and I believe the new results on out-of-distribution to be very valuable. Furthermore, I agree with AnonReviewer2 that the quality of the paper significantly improved. I will thus increase my score accordingly.

---

> > > ### Author Response · Authors · 2019-11-15
> > > **Thanks.**
> > >
> > > Thank you for suggesting those experiments, comments, responses, and for increasing the score.

---

### Official Review · AnonReviewer2 · 2019-10-23
**Official Blind Review #2**

**Rating:** 8

**Review:**

This paper proposes Meta Q-Learning (MQL), an algorithm for efficient off-policy meta-learning. The method relies on a simple multi-task objective which provides initial parameter values for the adaptation phase. Adaptation is performed by gradient descent, minimizing TD-error on the new validation task (regularizing towards initial parameter values). To make adaptation data efficient, the method makes heavy use of off-policy data generated during meta-training, by minimizing its importance weighted TD-error. Importance weights are estimated via a likelihood ratio estimator, and are also used to derive the effective sample size of the meta-training batch, which is used to adaptively weight the regularization term. Intuitively, this has the effect of turning off regularization when meta-training trajectories are “close” to validation trajectories. One important but somewhat orthogonal contribution of the paper is to highlight the importance of context in meta-learning and fast adaptation. Concretely, the authors show that a simple actor-critic algorithm (TD3), whose policy and value are conditioned on a context variable derived from a recurrent network performs surprisingly well in comparison to SoTA meta-learning algorithms like PEARL. MQL is evaluated on benchmark meta-RL environments from continuous control tasks and is shown to perform competitively with PEARL.

I have mixed opinions on this paper. On the positive side, and subject to further clarifications (see below), the paper seems to confirm that multi-task learning is almost sufficient to solve current meta-RL benchmarks in continuous control, without adaptation, as long as policy and critic are conditioned on a recurrent task context. This either highlights the strength of multi-task learning, or the inadequacies of current meta-RL benchmarks: either of which will be of interest to the community. On the other hand, the proposed MQL algorithm is only shown to significantly outperform this new baseline TD3-context agent on 1 of 6 tasks (Ant-Goal-2D), and furthermore the ablative analysis seems to suggest that the importance weighting and adaptive weighting of trust region are not very effective, and do not significantly change the performance of the method. The take-away seems to be that while context is crucial to generalization on these validation tasks, adaptation is not but can indeed be improved in a data-efficient manner with fine-tuning. MQL is only then a second thread to this story.


Clarifying Questions:

* The text and figures could be much clearer with respect to what is being measured/presented. Can you confirm that you report average validation returns as a function of meta-training steps, where validation performance is estimated after training on (at most) 2x200 transitions from the validation task (|D_new|=400)? This would match the protocol from PEARL. Fig 4(a) would then imply that one can get close to SoTA results on Ant-Fwd-Back + Half-Cheetah-Fwd-Back with no adaptation whatsoever (using TD3-context).
* Could the authors provide more details about how context is generated, in particular whether the GRU is reset on episode boundaries or not? If the recurrent context does span episode boundaries, are rewards being maximizing over a horizon greater than the episode length (similar to RL2)?
* How do you reconcile your result that a deterministic encoder is sufficient, compared to Figure 7 of PEARL which shows stochasticity is paramount for performing structured exploration during adaptation?
* Ablative analysis seems to suggest that off-policy learning plays a very minimal role during adaptation (beta=0). Can you confirm this interpretation? Would this not suggest that regularization via fine-tuning (regularized to multi-task prior) and context are sufficient to solve these meta-RL tasks? This would be a sufficient contribution by itself but unfortunately does little to validate the proposed method.
* Could you repeat the ablative analysis on all of the 6 tasks? Currently, this is performed on the two tasks for which TD3-context does best which leaves little room for improvement.
* Section 4.4. Propensity Score Estimation. “Use this model to sample transitions from the replay buffer that are similar to the new task”. Is this done via rejection sampling? This should be described more prominently in Section 3, along with a detailed description of the proposed MQL algorithm.

Detailed Comments:
* Paper desperately needs an algorithmic box, which clear and concise description of algorithm (how losses are interleaved, etc). Importantly, do the authors pretrain \theta_meta using the multi-task loss before doing adapation?
* Please add more informative labels to axes for all your figures: timesteps for [validation,training] ? Same for return. Please also augment captions to make figures as stand-alone as possible.
* MQL encompasses technique for off-policy training using a discriminator to estimate a likelihood ratio. It would be nice to evaluate this in standard off-policy learning setting, instead of it being limited to meta-learning.

**Experience Assessment:**

I have read many papers in this area.

**Review Assessment: Checking Correctness Of Derivations And Theory:**

I assessed the sensibility of the derivations and theory.

**Review Assessment: Checking Correctness Of Experiments:**

I assessed the sensibility of the experiments.

**Review Assessment: Thoroughness In Paper Reading:**

I read the paper thoroughly.

---

> ### Author Response · Authors · 2019-11-14
> **Response to Review #2 (Part 1)**
>
> Thank you for your feedback. Please also see the main comment above. We hope you will consider increasing your score after seeing our response.
>
> >> MQL is only shown to significantly outperform TD3-context on 1 of 6 tasks
> Yes, the difference between TD3-context (which is a baseline we have proposed) and MQL is minor for these meta-RL benchmark tasks. However the MQL is much better than TD3-context because of its explicit adaptation for out-of-distribution tasks. Please see the summary of the results of this experiment in the comment above and the full results in Appendix D of the updated paper.
>
> >>  furthermore the ablative analysis seems to suggest that the importance weighting and adaptive weighting of trust region are not very effective
> It depends on the task. For the Ant-Fwd-Back environment, where there is strong overlap between training and test tasks, the value of beta is almost one (Fig. 5). Due to this, MQL is much better (return ~1000 in Fig. 3) than TD3-context (return ~825 in Fig. 2) for this environment. A similar gap is also seen in the out-of-distribution experiment where beta ~ 0.2 throughout training. The key point is that the two terms are self-adapting to the problem, they kick in when necessary and disable themselves without hurting performance when the new task is quite different (Fig. 7 in Appendix D). The user does not need to tune them by hand; they are essentially free terms.
>
> >> take-away seems to be that while context is crucial to generalization on these validation tasks, adaptation is not.
> Context is necessary but not sufficient, it takes you most of the way on these tasks. Further, note that our GRU-based context variable does adapt the representation, it does not however use gradients. As the out-of-distribution experiment suggests, explicit adaptation is essential to perform well on more challenging tasks. So we would qualify the reviewer’s conclusion a bit: “our context-based adaptation is sufficient for the current meta-RL benchmarks”. Certainly more realistic benchmarks for meta-learning---the literature does not have them yet---would be ones that explicitly require meta-learning and strong adaptation.
>
> >> MQL is only then a second thread to this story
> There are two main points in our paper. First, the fact our context-based multi-task training takes gets a bulk of the performance suggests that meta-RL benchmarks need rethinking. Second, we have designed an algorithm for adaptation that is fundamentally sound and well-suited to off-policy learning. This is in contrast to on-policy meta-RL methods (MAML, ProMP) which cannot exploit old data. The out-of-distribution experiment validates our second contribution although this validation was not as strong in the initial draft. The reviewer notes that they find the first contribution sufficient.
>
> >> Make text and figures clearer. Can you confirm that you report average validation returns as a function of meta-training steps?
> We made the text and figures clearer. For all experiments, we exactly followed (Rakelley et al., have released their code and training logs) the evaluation protocol of PEARL. As the reviewer identifies, indeed, our experiments suggest that one can get close to SoTA results on some environments with no gradient-based adaptation whatsoever with our context-based adaptation.
>
> >> How is the context generated?
> We use a Gated Recurrent Unit (GRU) to create context. The GRU is reset after an episode finishes and is reinitialized to zero at the beginning of the next episode. The sequence length of the GRU is upper bounded by the episode length. The recurrent context does not span episode boundaries. We made this choice because we wanted the simplest possible context mechanism.

---

> > ### Author Response · Authors · 2019-11-14
> > **Response to Review #2 (Part 2)**
> >
> >
> > >> How do you reconcile your result that a deterministic encoder is sufficient, PEARL shows stochasticity is paramount.
> > Yes, these are conflicting observations with respect to the experiments of PEARL. We believe there are two main reasons for this. The experiment in PEARL that shows that stochasticity is paramount is only on a different sparse-reward environment, they do not show similar results for standard benchmark environments. We only did experiments on the standard environments to be consistent with the rest of the literature. Second, the two algorithms, namely PEARL and MQL are quite different. While it may be true that PEARL requires stochasticity of the encoder, as our experiments show, it is not necessary for MQL. MQL is therefore a simpler algorithm and yet works well on benchmark problems.
> >
> > >> Could you repeat the ablative analysis on all of the 6 tasks?
> > Good idea. Please see Appendix C for ablative analysis on two additional environments: Walker-2D and Half-Cheetah-Vel.
> >
> > >>  Is propensity score estimation done via rejection sampling?
> > No. We fit a propensity score estimator to the data and simply weigh the mini-batch of uniformly randomly chosen experiences from the replay-buffer by beta.
> >
> > >> Please add more informative labels to axes for all your figures: timesteps for [validation,training]
> > Done. The X-axis of figures is “meta-training time-steps”. The Y-axis is “average validation returns”.
> >
> > >> Paper desperately needs an algorithmic box
> > We have included the pseudo code in the updated version of the paper in Appendix A.
> >
> > >> MQL encompasses technique for off-policy training using a discriminator to estimate a likelihood ratio. It would be nice to evaluate this in standard off-policy learning setting, instead of it being limited to meta-learning.
> > This is a great idea, thanks. We will consider this for future work.

---

> > > ### Comment · AnonReviewer2 · 2019-11-14
> > > **Official Blind Review #2**
> > >
> > > Thank you to the authors for the thoughtful and thorough rebuttal.
> > >
> > > My main criticism of the paper was that despite a clear contribution, which highlighted the flaws of current meta-RL benchmarks, the efficacy of MQL itself over the baselines was not clearly established. The new experiments on out-of-distribution tasks however clearly paint a different story, with MQL outperforming PEARL and TD3-context significantly, precisely in the regime where you would expect their off-policy correction and adaptive regularization scheme to help. Hats off to AnonReviewer3 for recommending these experiments!
> > >
> > > In addition to the above, the paper seems much improved in terms of clarity (concise description of algorithm, captions, clarifying evaluation, etc) and controls (with extra ablative analysis). Taken all together I am now convinced this paper should be published at ICLR and am amending my score to reflect this.
> > >
> > > My only recommendation at this point is to move the new results from Appendix D into the main text, and expand on these using a broader range of environments.

---

> > > > ### Author Response · Authors · 2019-11-15
> > > > **Thanks**
> > > >
> > > > Thank you for your comments, response, and for increasing the score.
> > > > We will move that experiments into the main paper

---

### Author Response · Authors · 2019-11-14
**Response to all reviewers**

We thank the reviewers for their feedback that we've used to greatly improve the paper. We hope you will consider increasing your scores after seeing our responses.

We first summarize our response and the results of additional suggested experiments here. We have responded to the concerns of the reviewers as individual comments below.

All reviewers agree that our results are strong and our contributions are valuable to the community. To summarize our contributions: (i) we demonstrate that our multi-task learning formulation conditioned on task-specific context is sufficient to perform well on current meta-RL benchmarks and (ii) we devise a method to exploit past data from the meta-training replay buffer for improving the performance on the new task. The main concerns of the reviews are:

>> Reviewer 2 is concerned about whether MQL is better than our off-policy TD3-based baseline

To address this, we ran an experiment with out-of-distribution test tasks as suggested by Reviewer 3. We used the Half-Cheetah-Vel environment with disjoint training and test target velocities. Our results (Appendix D in the updated draft) show that explicit adaptation of MQL is essential to perform well in this situation. In contrast, both our TD3-based, context-dependent baseline and a recent state-of-the-art algorithm named PEARL perform significantly poorly on this task. This experiment demonstrates the importance of explicit off-policy adaptation on top of our simple multi-task training.

>> Reviewer 1 disagrees with the meta-learning narrative of our method

We have addressed a longer individual comment addressing this concern which we summarize here. Meta-training aims to train a policy on multiple tasks that can be adapted efficiently on a new task. The improved efficiency of a meta-learner comes from two things: (i) a better inductive bias to initialize the learning (Schmidhuber 97, Baxter 95, Baxter 97, Baxter 00 , Mitchell 80 etc.),  or (ii) creating a more efficient learning rule (Bengio et al., 97, Lee et al., 19 etc.). Meta-learning lacks a rigorous mathematical definition so it is difficult to argue as to what does not constitute meta-learning and what absolutely does. We, and most of the recent literature, e.g., MAML by Finn et al., 17, or Prototypical Networks by Snell et al., 17 use the notion (i) above, the one where a meta-learner builds a better inductive bias from the training tasks. The key difference between us and the literature is that our algorithm for meta-training is different from that for adaptation. We use our simple multi-task learning for training (as our experiments and recent trend in the computer vision literature shows, this is sufficient) and adapt for the new task using a combination of new data and past off-policy data.

We consciously take a very simplistic view of meta-learning. In an area rife with confusion we believe doing so is a virtue and a necessary first step towards understanding meta-learning. We have modified the narrative in the related work section to elucidate the above point more strongly.

We do not believe any of the above concepts bear resemblance to transfer learning (which starts from _one_ source task), domain randomization or feature reuse which the reviewer mentions.

RL algorithms---meta-RL, especially so---are notorious for their fragility and complexity. Our work demonstrates a lacuna in existing meta-RL benchmarks as well as a new, simple algorithm for off-policy meta-RL. The former is crucial in a burgeoning field like meta-RL. The latter is a fundamentally sound and efficient algorithm that we show to work well on out-of-distribution tasks and is likely to retain its strong performance on better meta-RL benchmarks. We are in vehement agreement with all the reviewers that even the first finding is surprising and valuable; we will polish the narrative to focus on it better. The insufficiency of existing benchmarks to measure the full impact of our second contribution should not take away its importance and novelty.

[Schmidhuber et al., 97]  Shifting inductive bias with success-story algorithm, adaptive levin search, and incremental self-improvement.
[Finn et al., 17] Model-agnostic meta-learning for fast adaptation of deep networks.
[Snell et al., 17] Prototypical networks for few-shot learning.
[Bengio et al., 95]  On the optimization of a synaptic learning rule.
[Lee et al., 19] ]Meta-learning with differentiable convex optimization
[Baxter, 95] Learning internal representations
[Baxter, 97] A Bayesian/Information Theoretic Model of Learning to Learn via Multiple Task Sampling.
[Baxter, 00] A model of inductive bias learning.
[Mitchell, 80 ] The need for biases in learning generalizations.

---

### Decision · Program_Chairs · 2019-12-19

**Decision:**

Accept (Talk)

**Comment:**

This paper’s contribution is twofold: 1) it proposes a new meta-RL method that leverages off-policy meta-learning by importance weighting, and 2) it demonstrates that current popular meta-RL benchmarks don’t necessarily require meta-learning, as a simple non-meta-learning algorithm (TD3) conditioned on a context variable of the trajectory is competitive with SoTA meta-learning approaches.

The reviewers all agreed that the approach is interesting and the contributions are significant. I’d like to thank the reviewers for engaging in a spirited discussion about this paper, both with each other and with the authors. There was also a disagreement about the semantics of whether the approach can be classified as “meta-learning”, but in my opinion this argument is orthogonal to the practical contributions. After the revisions and rebuttal, reviewers agreed that the paper was improved and increased their ratings as a result, with all recommending accept.

There’s a good chance this work will make an impactful contribution to the field of meta-reinforcement learning and therefore I recommend it for an oral presentation.